# Use of Wing Geometric Morphometric Analysis and mtDNA to Identify Africanization of *Apis mellifera* in the Central Highlands of Ecuador

**DOI:** 10.3390/insects15080628

**Published:** 2024-08-20

**Authors:** Diego Masaquiza, Lino Curbelo Rodríguez, José Zapata, Joffre Monar, Maritza Vaca, Leonardo Porrini, Martin Eguaras, Martin Daniele, Dora Romero, Amilcar Arenal

**Affiliations:** 1Sede Orellana, Escuela Superior Politécnica de Chimborazo, El Coca 220150, Ecuador; jose.zapata@espoch.edu.ec (J.Z.); jmonar@espoch.edu.ec (J.M.); 2Center for Animal Development and Production Studies, Ignacio Agramonte Loynaz University of Camagüey, Camagüey 74650, Cuba; lino.curbelo@reduc.edu.cu; 3Facultad de Ciencias Pecuarias, Escuela Superior Politécnica de Chimborazo, Riobamba 060106, Ecuador; maritza.vaca@espoch.edu.ec; 4Department of Biology, Faculty of Exact Sciences, National University of Mar del Plata, Mar del Plata 7600, Argentina; leoporrini@gmail.com (L.P.); mjeguaras@gmail.com (M.E.); 5Sede Alto Valle y Valle Medio, Escuela de Veterinaria y Producción Agroindustrial, Universidad Nacional de Río Negro, Choele Choel 8360, Rio Negro, Argentina; martindaniele@gmail.com; 6Laboratorio de Parasitología, Unidad de Diágnostico, Torreón del Molino, Facultad de Medicina Veterinaria y Zootecnia, Universidad Veracruzana, Veracruz 91697, Mexico; dromero@uv.mx; 7Department of Infectious Diseases and Pathology, School of Veterinary Medicine, St. Nicholas University, Morne Daniel, Roseau 00109-8000, Dominica; amilcar.arenal@gmail.com

**Keywords:** *Apis mellifera scutellate*, Africanization, altitudinal floors, morphotypes, haplotype

## Abstract

**Simple Summary:**

The presence of the Africanized bee on the American continent allowed hybridization processes, which led to a high genetic diversity. However, distinct morphotypes with unique characteristics emerge at different altitudinal levels across the continent. Ecuador possesses high biodiversity and offers ideal conditions for Africanized hybrid development. Nevertheless, it is crucial to identify the formation of morphotypes adapted to the area in order to propose conservation plans for these insects, since individuals with excellent behavior in both hygiene and defensiveness, desirable traits in Africanized bees, have been observed previously.

**Abstract:**

Seventy-five samples were collected from 15 beehives in the central highlands of Ecuador (Tungurahua–Chimborazo) to assess Africanization in managed bee populations using wing geometric morphometric and mitochondrial DNA analyses. The results indicated that when grouping the apiaries based on altitudinal floors into 2600–2800, 2801–3000, and 3001–3274 m above sea level, differences (*p* < 0.001) were observed. The morphotypes were similar in the first two floors, but the third indicated that altitude plays a crucial role in the differentiation of populations. When comparing with the pure subspecies, we found differences (*p* < 0.001); the nearest Mahalanobis distance was for *Apis mellifera scutellata* (D2 = 3.51), with 95.8% Africanization via father in the area. The maternal origin of all patterns belonged to lineage A (*A. m. scutellata*), with seven haplotypes. The most frequent haplotypes were A26 and A1; however, the A1q haplotype was not detected at the national level or in nearby countries. The identified haplotypes do not coincide with A4, which is predominant in South Africa and Brazil. The results indicate a double origin due to their presence in North Africa and the Iberian Peninsula. The formation of specific morphological groups within ecoregions is suggested.

## 1. Introduction

The conservation and sustainable use of biodiversity for food and agriculture play a crucial role in the fight against hunger to ensure environmental sustainability and increase agricultural and food production [1]. The decrease in species worldwide caused by different factors [2] significantly alarms researchers and producers. As a result, researchers have orientated their efforts toward characterization studies, technological exploitation, and sustainable use of species.

The identification and classification of *A. mellifera* have a long history, with various methods employed since ancient times. Early classifications of *A. mellifera* relied on morphological and behavioral characteristics, often linked to the geographic distribution of the observed specimens [3]. These included traditional morphometry [4,5], allozyme and isozyme analysis [6,7], nuclear DNA molecular markers [8], mitochondrial DNA [9,10,11], microsatellites [12,13,14], cuticular hydrocarbons [15], and single-nucleotide polymorphisms (SNPs) [16].

In recent years, geometric morphometry (GM) has taken on great importance and is the most widely used tool for identifying subspecies [17]. GM has emerged as a valuable tool for *A. mellifera* identification due to its high accuracy in discriminating between species, subspecies, and hybrids [18,19]. Some studies suggest that geometric morphometrics might sometimes be even more effective than molecular markers for subspecies identification within *A. mellifera* [20].

Molecular markers such as mitochondrial DNA can help validate and expand our knowledge of species biodiversity [21]. The genetic diversity within a population represents the raw material for adaptation, allowing organisms to evolve and better cope with changing environments and demands [22]. Information on the origin and history of animal genetic resources underpins their sustainable management [23]. Prior evaluation of genetic variability and gene flow within populations is crucial for effective gene pool management in animal genetic resources. This assessment helps establish appropriate selection programs for breeding and conservation efforts [24].

In Brazil, in 1956, after an attempt to improve European bees, to obtain a hybrid product of interbreeding between European and African subspecies with good temperament and searching habits [25], a swarm escape occurred, producing a hybridization with native bees [26]. The continent’s extensive geographic distribution and environmental variation have fostered micro-evolutionary adaptations in hybrid populations [27]. Perhaps due to their remarkable genetic plasticity and adaptability, these hybrids have formed and dispersed across diverse morphoclimatic patterns, acquiring unique characteristics [28]. Africanized bees are present from northern Argentina to the southern and central United States [11] and occupy a range of approximately 20 million km^2^. Their high colonizing capacity has led to one of the fastest and most rapid biological invasions on record [29,30].

The introduction of African honey bees (*Apis mellifera scutellata*) through hybridization has impacted natural ecosystems and beekeeping activities on the continent. Their defensive behavior has posed a particular challenge. The Africanization process began in Ecuador during the 1970s [31], rapidly spreading throughout the country and causing significant damage to Ecuadorian beekeeping.

In the highlands of Ecuador, altitude may be a key factor influencing the variability of bee populations, particularly in continental Africanization. Distinguishing bee subspecies or lineages is crucial for two main reasons. First, it aids in conserving bee biodiversity. Second, it allows us to assess the extent of Africanization, serving as a baseline for designing strategies to conserve identified genotypes.

Mitochondrial DNA analysis is a powerful tool to identify bee species and trace their maternal origin. Additionally, morphometric analysis of traits can be a good indicator of nuclear introgression, providing insight into the paternal genetic contribution [32]. Studies have identified five distinct lineages of bees [33], likely due to variations in climate and flora caused by glacial and post-glacial periods [34].

Therefore, this study aims to measure the Africanization of the honey bee with geometric morphometry and mitochondrial DNA in the Ecuadorian highlands at altitudes between 2600 and 3274 m above sea level.

## 2. Materials and Methods

### 2.1. Sampling

To collect samples for analysis, we visited 15 apiaries (beekeeping establishments) in the Ecuadorian provinces of Tungurahua and Chimborazo (Figure 1). This territory presents the particularity of being traversed from north to south by the mountainous system of the Andes. The climate of the center area of Ecuador can range from temperate semi-wet to humid. It is warm and dry in the valleys and high cold mountain conditions on the paramos, over 3400 m above sea level. The temperature is linked to height (i.e., between 1500 and 3000 m.a.s.l.) [35]. Seven hundred and fifty samples were collected, focusing on the largest apiaries.

The hives that we worked with were commercial and were selected according to the characterization of the beekeepers [36]; inclusion and exclusion criteria were considered regarding the research goals.

This study encompassed apiaries utilizing Langstroth hives, colonies demonstrating robust health (evidenced by the occupation of seven combs, including an average of three brood combs, aligned with [37]; honey production exceeding the national average of 10.2 kg per hive [38]; and a history of queen stability. Conversely, exclusion criteria were implemented for swarm hives, apiaries practicing transhumance, and beekeepers declining participation in the research.

Of the eighteen apiaries, three were excluded: two in the province of Tungurahua (for swarming and transhumance) and one in Chimborazo (refusal of the beekeeper). The hives under study had a breeding chamber and two half honey supers.

An average of 250 worker bees from the central combs of the brood chamber were taken per hive, which guaranteed variation between the colonies [39], in five hives per apiary, and were stored in 90% ethanol in a freezer (−20 °C) until examination.

### 2.2. Geometric and Morphometric Qualities

The left forewing was dissected from 10 bees per colony from the 15 apiaries (750 wings) [39]. The wings were mounted in glass photographic frames and scanned with a PlusteK OpticFilm 8100 (7200 dpi). Nineteen homologous landmarks [40], corresponding to specific vein intersections on the wing, were manually identified in the obtained images (Figure 2). This process was facilitated by tpsDig2 v2.16 software, which generated TPS files using tpsUtil v1.46 software [41].

The analysis additionally included 50 images of the left forewing for each documented pure subspecies known to have been introduced into the country, *A. m. carnica*, *A. m. ligustica*, *A. m. mellifera*, and *A. m. scutellata*, obtained from the Morphometric Bee Data Bank in Oberursel, Germany.

To the files with the coordinates of the landmarks, a Procrustes adjustment was applied (to eliminate the variation caused by the differences in the size, position, and orientation of the wings) to align and obtain the coordinates of the centroid. The overlapping coordinates were projected in a shaped space tangent [42], thus providing the shape parameters used in multivariate analyses. Based on the spectral decomposition of the covariance, principal component analysis (PCA) was carried out as an exploratory analysis with all the measurements of the study area. The differences in shape between apiaries were obtained by calculating the Mahalanobis distance (degree of separation between populations) by canonical variable analysis (CVA) using the MORPHOJ package [43].

A discriminant function analysis (DFA) allowed the identification of each sample. A cross-validation test verified the reliability of the data, and a permutation test was carried out for all pairwise tests.

First, the entire dataset was analyzed to identify potential natural groupings. Second, bees were grouped based on altitude. A bivariate correlation (SPSS version 21) between wing measurements (centroid) and altitude was performed. This analysis resulted in three altitudinal groupings: 2600–2800 m above sea level (m.a.s.l.), 2801–3000 m.a.s.l., and 3001–3274 m.a.s.l. Finally, individual bees were classified at the subspecies level. This analysis compared the morphometric variation patterns of the studied bees with known pure subspecies to assess the extent of Africanization.

A UPGMA cluster [44] was made with the Mahalanobis distances obtained from morphometric data to show the clustering among honey bee populations, with the pairwise distance method, using the MEGA 7 software [45]; placebo samples of bees *Tetragonisca angustula* were used.

### 2.3. Mitochondrial DNA

Genomic DNA was extracted directly from the thorax of 15 workers (one per apiary) using the alkaline lysis method. This method relies on cell lysis to release DNA without further purification [46]. Each sample was immersed in 50 µL of alkaline lysis solution (25 mM NaOH, 0.2 mM EDTA, pH 12) and incubated at 95 °C for 30 min. Subsequently, the samples were cooled to 4 °C in a Mastercycler ^TM^ thermocycler (Eppendorf, Germany) and were added to 50 µL of neutralizing solution (40 mM Tris-HCl, pH 5). DNA samples were frozen at −20 °C until analysis.

### 2.4. DNA Amplification

Genomic DNA was used to amplify the intergenic region tRNALeu-COII with the primers E2 (5′- GGCAGAATAAGTGCATTG-3′) and H2 (5′-CAATATCATTGATGACC-3′). Amplification reactions were performed in 50 μL solution containing 1× DreamTaq Buffer, 1.5 mM MgCl_2_, 1 U of Taq Polymerase, 2 μL of the sample, 0.6 μM of each primer and 0.2 mM of dNTP. The PCR program followed the Garnery, Solignac [47] protocol: initial denaturation at 94 °C for 5 min followed by 40 cycles each at 94 °C for 1 min, 55 °C for 45 s, and 72 °C for 1 min, and a final extension step at 72 °C for 10 min. The size of the amplified fragment was verified by electrophoresis in 1% agarose gel.

### 2.5. Sequencing and Phylogenetic Analysis

The samples that amplified successfully were purified and sequenced with the forward primer (E2) used in the PCR protocol. The molecular analysis sequences were assembled, edited, and analyzed with the Molecular Evolutionary Genetics Analysis 7 (MEGA 7) program. Subsequently, sequences were compared with the NCBI Genbank database using the BLAST algorithm (Basic Local Alignment Search Tool) to determine the sequence(s) with the highest similarity. A phylogenetic tree was constructed with the sequences aligned with the maximum likelihood method.

## 3. Results

In the analysis of the sampled honey bees, when studying the apiaries as a whole, no specific groups were formed where the first 13 canonical variables were necessary to explain 80% of the total variation between the colonies, suggesting the existence of intense gene flow between colonies (Figure 3A). Moreover, there was a highly significant negative correlation (r = −0.32 **) between altitude and centroid size; these results indicate that at higher altitudes, the size of the wing shape tends to be smaller, which may be associated with the effect of altitude in addition to the Africanization process.

Grouping apiaries based on their altitude in the highlands (2600–2800 m above sea level (m.a.s.l.), 2801–3000 m.a.s.l., and 3001–3274 m.a.s.l.) revealed significant morphometric variations (*p* < 0.001). While no morphometric differences were observed between the two lower altitude levels (2600–2800 m.a.s.l. and 2801–3000 m.a.s.l.), significant variations were detected between these and the highest altitude level (3001–3274 m.a.s.l.) (Figure 3B). The total variation in the sample (100%) was represented by the first two canonical variables. These results indicate significant morphometric differentiation (*p* < 0.001) between the sampled honey bees studied and all the pure subspecies examined.

Likewise, the CVA of the data grouping confirmed the differentiation of the pure subspecies of the highland populations; the scatter plot based on the 19 landmarks showed the maximum separation between the groups (Figure 3C). Notably, *A. m. scutellata* overlaps with samples from all apiaries, suggesting a potential influence and the formation of novel morphotypes in the highland bee population.

The CVA identified that the first six canonical variables explained 80% of the total variation between the four groups of *A. mellifera* and the individuals under study. According to the discriminant analysis, 94.74% of the individuals were classified in their respective groups. However, when applying the cross-validation test, 91.79% of them were correctly identified.

A total of 90.2% of individuals were classified as Africanized, 5.7% as *A. m. scutellata*, 1.8% as *A. m. ligustica*, 1.2% as *A. m. carnica*, and 1.1% as *A. m. mellifera* (Table 1). However, the frequency of Africanized morphotypes did not differ for each province.

The degree of Africanization was similar, between 90% and 95.8%, in the three altitudinal floors (Table 2).

The Mahalanobis distances were closer for *A. m. scutellata* (D2 = 3.51); however, longer distances were seen for the subspecies *A. m. ligustica* (D2 = 4.28), *A. m. carnica* (D2 = 4.63), and *A. m. mellifera* (D2 = 4.73). The dendrogram showed three main clusters (Figure 4).

The first cluster encompassed individuals of the M lineage (*A. m. mellifera*), the second cluster comprised members of the C lineage (*A. m. ligustica* and *A. m. carnica*), and the third cluster consisted of two sub-groups: populations of the A lineage (*A. m. scutellata*) and local Africanized hybrids. A nine-locus phenogram effectively differentiated the three lineages and the study subjects.

Intergenic regions of the RNAtleu-COll showed polymorphisms in sequence and size among individuals. The amplification of the RNAtleu-COll fragment resulted in bands with fragments of lengths of between 400 and 740 base pairs.

Samples were related to more than one sequence, and the haplotypes were already described; however, the lineage classification did not differ in any case. The maternal origins of all patterns were traced to Africa (lineage A), with seven distinct haplotypes identified. Notably, the A1q haplotype was detected, a variant previously unreported at the national or regional levels (Table 3). Genetic similarity among the observed haplotypes, assessed by shared nucleotide sites, ranged from 94.14% to 100%.

The most frequent detections were African haplotypes A26 and A1; the rest were detected less frequently, which indicated that the maternal Africanization process occurred throughout the area under study. The two percent covers all the nodes of the sequences of the intergenic region RNAtleu-COll of the mitochondrial DNA of the bee samples, which represents a high level of relation between the sequences in comparison (Figure 5).

## 4. Discussion

According to the criteria of Klok and Harrison [48], altitude can influence the size and shape of the structures of organisms because, as it is higher, the oxygen concentration in the air decreases and causes hypoxia, which can alter the development of individuals, making them smaller, which shows that it plays a crucial role in the differentiation of populations. This behavior could be because the migration of Africanized bees to high areas (above 2600 m.a.s.l.) is probably slower, which is fundamentally because the temperatures are less favorable for their development, and adaptation is more difficult than in other scenarios in the region.

The arrival of Africanized honey bees in Ecuador likely began in the lowland tropics, where the climate is more favorable. Cooler temperatures in the highlands likely presented a selection pressure, delaying colonization by Africanized honey bees until they could adapt. We observed that the highest altitude level differed from the preceding two. This finding aligns with the lower Africanization percentage of individuals at higher altitudes. In the highlands of Mexico, Medina-Flores, Guzmán-Novoa [29] reported 14% Africanization at altitudes between 1800 and 2400 m.a.s.l.; 44% between 1200 and 2000 m.a.s.l., and 69% between 1000 and 1400 m.a.s.l. This suggests that European honey bee genotypes are better maintained at higher altitudes with cooler climates.

The first hives introduced in Ecuador were of the honey bee ecotype *A. m. ligustica* [49], in the 1950s. The arrival of Africanized hybrids meant the abrupt displacement of the populations of these bees due to competition. Likewise, Wallberg, Han [50] argued that the success of Africanized bees in spreading is related to the presence of genes with possible functions regarding the motility and maturation of spermatozoa, for which reason bees in South America have, to a large extent (70–90%), lineage of African origin.

The most common haplotypes in nearby countries in the region were A1, A4, and A26; we know that in Uruguay, Central–South Brazil, and Venezuela, the most common is A4 [51], and in Colombia, the most common are A1e, A26a, A1, A4, A26d, and A26c [52]. In northern Ecuador, diversity related to lineage A and C was found: A26a, A26c, A30, A1, A1e, A31, seq1, seq2, C1, C2j, and C31 [53].

The high presence of the A1 haplotype is consistent with the existence of a negative correlation between latitude and the presence of the haplotype, with a higher frequency of A1 towards the north of Brazil [54]. It can also be noted that there is a similar frequency of haplotypes across the study area; these results agree with the morphometric analyses (96% paternal Africanization), so it can be asserted that the Africanization process in the highlands has both a matrilineal and patrilineal origin (it is bidirectional). More recently, Tibatá, Arias [52] confirmed that there are 98.3% of populations with African mitotypes. However, a prevalent practice in this region—capturing wild swarms to increase hive numbers—might inadvertently contribute to the spread and survival of Africanized bees [36].

Acosta [53] suggests an Africanization towards the north of Ecuador, with a high presence of haplotypes of the A lineage and a reduction as it enters the central part of the country. The present study suggests that Africanization may also have occurred in the eastern region, following an adaptation period that allowed bees to migrate to different altitudes. In this study, the apiaries showed a haplotype from the A lineage, with the first report of the A1q haplotype at the national level.

Based on morphometric analyses, the Africanization process in the highlands is still in progress. However, the complete Africanization of the country is limited due to the introduction of imported European queens, as pointed out by Branchiccela, Aguirre [30] in Uruguay.

The haplotypes identified in the present investigation do not coincide with A4, the predominant haplotype in southern Africa [55], Brazil, and Uruguay [54]. Haplotype A patterns have also been reported in North Africa and the Iberian Peninsula [55]. Therefore, it could have a double-origin in America, either from the Iberian Peninsula or the introduction of *A. m. scutellata* in Brazil in 1956. For this reason, the use of new polymorphic genes is necessary to identify the asymmetry in gene flow.

The haplotype variability shows a high gene flow among the bees of the studied region, which delays the formation of morphotypes as an adaptive response of these insects to the specific conditions of each altitudinal floor in the highlands of Ecuador. Additionally, the disorganized introduction of queens of different subspecies into the country by beekeepers also delays the formation of morphotypes.

The presence of Africanization in *Apis mellifera* in the central highlands of Ecuador, with a diversity of haplotypes of the A lineage, indicates a constant and dynamic exchange of germplasms. In addition, the existence of specific morphological groups within the ecoregions suggests a process of adaptation in bees to the environmental conditions of the highlands.

However, further analysis is needed to assess the impact of meteorological variables on the dispersion of Africanization. Moreover, it is necessary to utilize new polymorphic genes to identify the asymmetry of gene flow in *Apis mellifera* from the central highlands of Ecuador. This will enable the assessment of genetic diversity for incorporation into genetic programs.

Guzmán-Novoa, Correa [56] pointed out that the first cause of the displacement of European genes via maternal inheritance is the usurpation of the colonies by queens. However, Mortensen and Ellis [54] indicate that matrilineal usurpation is not the principal contributing cause to African haplotypes, but rather, hybridization events are.

## 5. Conclusions

The presence of Africanization in *Apis mellifera* in the central highlands of Ecuador, with a diversity of lineage A haplotypes, indicates a constant and dynamic germplasm exchange. The formation of specific morphological groups within ecoregions suggests a process of adaptation of bees to the environmental conditions of the highlands. Future research should prioritize evaluating the impact of meteorological variables on the dispersal patterns of Africanized honey bees in Ecuador. Such insights are crucial for devising effective conservation and management strategies for Ecuadorian honey bee populations.

## Figures and Tables

**Figure 1 insects-15-00628-f001:**
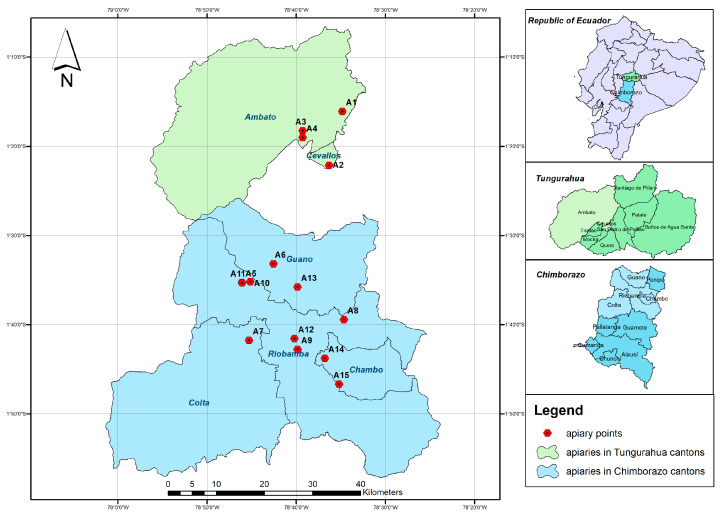
Map of the provinces of Tungurahua and Chimborazo with the geographical locations of the apiaries marked.

**Figure 2 insects-15-00628-f002:**
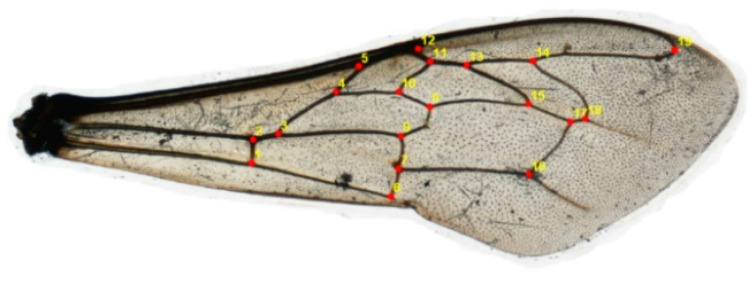
Location of 19 landmarks at venal intersections of the left forewing of *A. mellifera*.

**Figure 3 insects-15-00628-f003:**
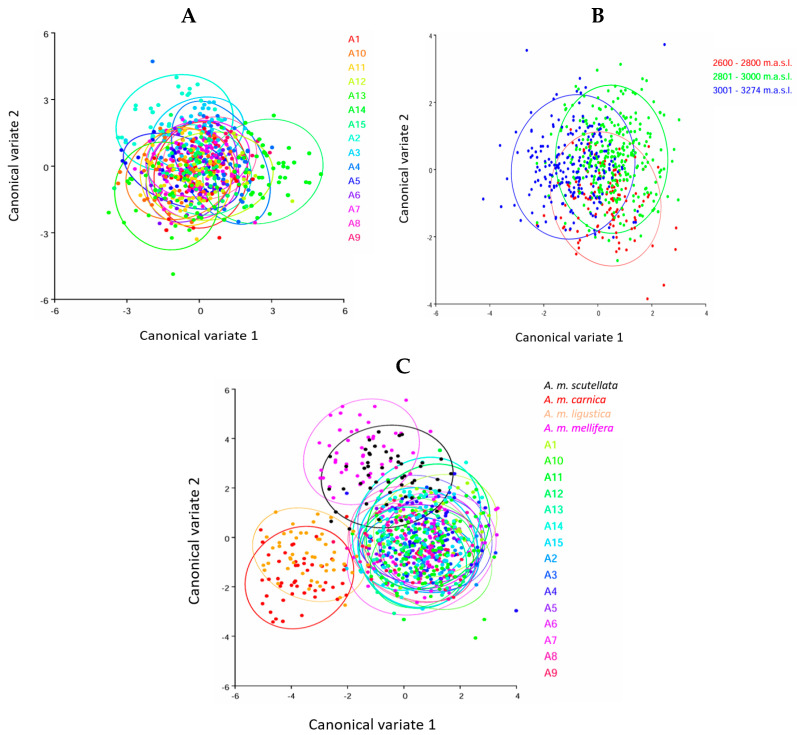
Scatter plot according to the landmarks of the left forewing of *Apis mellifera* from the central highlands of Ecuador. CVA: (**A**) considering the colonies as a whole and (**B**) considering three altitudinal floors (red, 2600–2800 m.a.s.l.; green, 2801–3000 m.a.s.l.; light blue, 3001–3274 m.a.s.l.). (**C**) considering pure subspecies (red, *A. m. carnica*, purple, *A. m. ligustica*; blue, *A. m. mellifera*; black, *A. m. scutellata* and apiaries under study A1–A15). Ellipses are drawn with 90% probability.

**Figure 4 insects-15-00628-f004:**
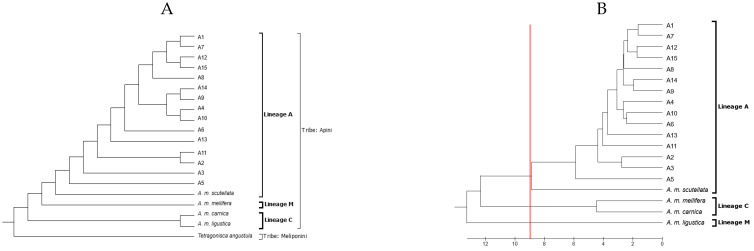
Dendrograms showing (**A**) groupings of pure subspecies of *Apis mellifera*, apiaries, and a placebo; (**B**) groupings of pure *Apis mellifera* subspecies and study apiaries of the central highlands of Ecuador, based on the Mahalanobis distances. Method: pairwise distance. Bee lineages: A, M, and C. Subspecies: *A. m. scutellata*, *A. m. mellifera*, *A. m. carnica*, and *A. m. ligustica*. Apiaries under study: 1–15. Red line = phenom. Placebo samples = *Tetragonisca angustula*.

**Figure 5 insects-15-00628-f005:**
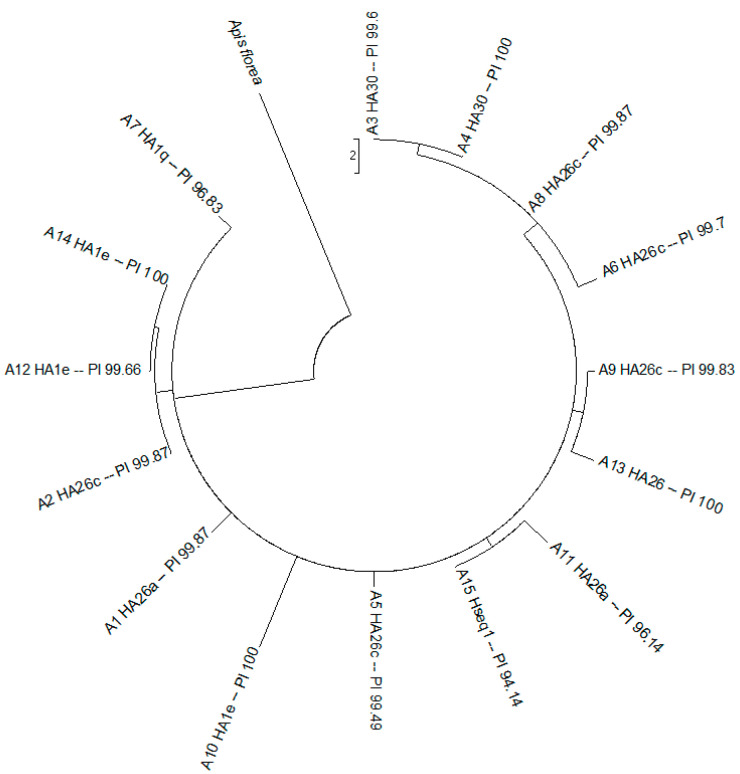
Phylogenetic tree of the sequences of the intergenic region RNAtleu-COll of the mitochondrial DNA of samples of bees from the central highlands of Ecuador. Method: maximum likelihood. analysis and visualization were performed in MEGA 7 software. Placebo sample: *Apis florea*. Study samples: A1–A15. H: haplotype. PI: percent identity.

**Table 1 insects-15-00628-t001:** Frequency of bees (wings) in the central highlands of Ecuador based on the pure subspecies of *Apis mellifera*.

Known Classification	N (Wings)	*A. m. carnica* (%)	*A. m. ligustica* (%)	*A. m. mellifera* (%)	*A. m. scutellata* (%)	AB* (%)
*A. m. carnica*	50	90	10	0	0	0
*A. m. ligustica*	50	10	90	0	0	0
*A. m. mellifera*	50	0	2	98	0	0
*A. m. scutellata*	50	0	0	0	94	6
AB^*^	750	1.2	1.8	1.1	5.7	90.2

AB* = Africanized bees, samples from the 15 apiaries.

**Table 2 insects-15-00628-t002:** Frequency of *Apis mellifera* morphotypes based on altitudinal floors in the central highland of Ecuador.

Morphotype (%)	Altitudinal Floors (m.a.s.l.)		
	2600–2800	2801–3000	3001–3274
	n = 15	n = 35	n = 25
Africanized	95.8	93	90
European	4.2	7	10

**Table 3 insects-15-00628-t003:** Similarity, haplotypes, and lineages in the analysis of mitochondrial DNA of representative samples of 15 apiaries of *Apis mellifera* from the central highland of Ecuador.

Apiary	Bee	Identity (%)	Haplotype	Lineage
A1	1	99.87	A26a	A
A2	2	99.87	A26c	A
A3	3	99.6	A30	A
A4	4	100	A30	A
A5	5	99.49	A26c	A
A6	6	99.70	A26c	A
A7	7	96.83	A1q	A
A8	8	99.87	A26c	A
A9	9	99.83	A1e	A
A10	10	100	A1e	A
A11	11	96.14	A26a	A
A12	12	99.66	A1e	A
A13	13	100	A26	A
A14	14	100	A1e	A
A15	15	94.14	Seq1	A

## Data Availability

Research data, including coordinates of the reference points and wing images, are shared in the public access repository “Zenodo”: Masaquiza, D., & Arenal, A. (2024). Collection of images and raw coordinates of honey bee (*Apis mellifera*) wings from the central highlands of Ecuador. Zenodo. https://doi.org/10.5281/zenodo.13340594, accessed on 19 August 2024.

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
