# Peer review of "Use of Wing Geometric Morphometric Analysis and mtDNA to Identify Africanization of Apis mellifera in the Central Highlands of Ecuador"

_insects, 2024, doi:10.3390/insects15080628_

Round 1
Reviewer 1 Report
Comments and Suggestions for Authors
Dear Authors,
Some corrections and suggestions have been made to the text of the article, attached.
I would also recommend that the description of the wing samples obtained from the databank be corrected in the materials and methods section (line 127). It seems like a typo, but it's important.
I especially recommend correcting the placebo sample description in Figure 4.
Best regards,

Author Response
- Some corrections and suggestions have been made to the text of the article, attached.
All corrections have been made
- Details such as last requeening, commercial queen use, feral swarms, or commercial colony use are more important.
Done
- This sentence should be reinterpreted due to overlapping samples. You can use the classification rates of the samples into the original groups with discriminant analysis
This was rewritten and discussed; however, we note that the higher altitude level differs from the previous two. This finding is consistent with the lower percentage of Africanization of individuals at higher altitudes.
- I would also recommend that the description of the wing samples obtained from the databank be corrected in the materials and methods section (line 127). It seems like a typo, but it's important.
Done
- I especially recommend correcting the placebo sample description in Figure 4.
Done
Reviewer 2 Report
Comments and Suggestions for Authors
Dear Authors,
I have carefully read your manuscript entitled "Use of wing geometric morphometric analysis and mtDNA to identify Africanization of Apis mellifera in the central highlands of Ecuador."
The article aims to identify Africanization of honey bees in the highlands of Ecuador using geometric morphometric analysis of wings and mitochondrial DNA analysis. A total of 75 samples were collected from 15 apiaries in the provinces of Tungurahua and Chimborazo, divided into three altitudinal levels ranging from 2600 to 3274 meters above sea level. The results indicate significant morphotype differences correlated with altitude, with a high percentage of Africanization (95.8%) attributed to the subspecies Apis mellifera scutellata.
The article addresses a timely and relevant topic, as Africanization of bees has a significant impact on beekeeping and local ecosystems. The combination of geometric morphometric techniques and mitochondrial DNA analysis represents a robust and innovative approach to studying genetic and morphological diversity in honey bees.
Below are my observations/suggestions that should be considered to enhance your work:
- Lines 107-108: The description of criteria for selecting apiaries and colonies is not sufficiently detailed. It is not specified whether there were specific inclusion or exclusion criteria or how apiaries were chosen. I suggest integrating this information.
- Lines 156-157: Specify the number of specimens considered for genetic analysis.
- Line 175: Specify how many sequences were obtained from sequencing. It is also important to know if sequencing of each amplified DNA sample was performed in both forward and reverse directions (F and R).
- Line 178: The obtained sequences should be deposited in a database, and accession numbers should be provided in the text. I suggest the authors include this information.
- Line 239: I suggest including an outgroup species in the dendrogram construction.
- Line 268: The discussion of practical implications of the results for beekeeping and biodiversity conservation is limited. I recommend expanding the discussion on practical implications, including considerations on the impact of Africanization on local beekeeping, possible management measures, and implications for the conservation of honey bee populations.
- The conclusions should also be reviewed and expanded as they appear to be somewhat inconsistent.
- Overall, some sections of the text could benefit from greater clarity and structure. Particularly, the transition between presenting results and discussing them is not always smooth. I suggest reorganizing some parts of the text to improve coherence and clarity, ensuring that the transition from results to discussion is clear and well-structured.
Please consider these suggestions to enhance the coherence, clarity, and completeness of your manuscript. Your study addresses an important issue, and refining these aspects will strengthen its impact and contribution to the field.
Comments on the Quality of English LanguageModerate editing of English language required.
Author Response
- Lines 107-108: The description of criteria for selecting apiaries and colonies is not sufficiently detailed. It is not specified whether there were specific inclusion or exclusion criteria or how apiaries were chosen. I suggest integrating this information.
The following inclusion and exclusion criteria were included
- Lines 156-157: Specify the number of specimens considered for genetic analysis.
Done
- Line 175: Specify how many sequences were obtained from sequencing. It is also important to know if sequencing of each amplified DNA sample was performed in both forward and reverse directions (F and R).
Done, It was only forward not assembling sequence
- Line 178: The obtained sequences should be deposited in a database, and accession numbers should be provided in the text. I suggest the authors include this information.
In process, we are waiting for the final number
- Line 239: I suggest including an outgroup species in the dendrogram construction.
A placebo sample of native bees (Tetragonisca angustula) of the meliponini tribe was introduced to the study.
- Line 268: The discussion of practical implications of the results for beekeeping and biodiversity conservation is limited. I recommend expanding the discussion on practical implications, including considerations on the impact of Africanization on local beekeeping, possible management measures, and implications for the conservation of honey bee populations.
Done
- The conclusions should also be reviewed and expanded as they appear to be somewhat inconsistent.
Done
- Overall, some sections of the text could benefit from greater clarity and structure. Particularly, the transition between presenting results and discussing them is not always smooth. I suggest reorganizing some parts of the text to improve coherence and clarity, ensuring that the transition from results to discussion is clear and well-structured.
The sessions were reorganized and the English was improved, being revised by 3 native speakers, which consequently improved the transition between paragraphs and sessions.
Round 2
Reviewer 2 Report
Comments and Suggestions for Authors
Heartfelt congratulations to the authors! I am thoroughly impressed with the significant enhancements made to the manuscript. Your hard work and dedication have truly paid off, resulting in a remarkable piece of work. Wishing you continued success in all your future endeavors.
All the best!